# Visible Light-Induced Cascade Sulfonylation/Cyclization to Produce Quinoline-2,4-Diones under Metal-Free Conditions

**DOI:** 10.3390/molecules28073137

**Published:** 2023-03-31

**Authors:** Yan Zhang, Ge Qiu, Fei Liu, Dongyang Zhao, Miao Tian, Kai Sun

**Affiliations:** 1School of Chemistry and Chemical Engineering, Henan Normal University, Xinxiang 453007, China; 2College of Chemistry and Chemical Engineering, Yantai University, Yantai 264005, China

**Keywords:** visible light, quinolone, radical, cyclization, *N*-heterocycles

## Abstract

A general visible light-induced sulfonylation/cyclization to produce quinoline-2,4-diones was achieved under photocatalyst-free conditions. The reactions were performed at room temperature, and various substituents (halogen, alkyl, aryl) and substituted products were obtained with 29 examples within 2 h. Large-scale synthesis and derivatization study via carbonyl reduction to produce easily modified hydroxyl groups and convenient *N*-Ts deprotection showed the potential utility of this strategy.

## 1. Introduction

The derivatives of quinoline-2,4-diones have shown unique biological activities in natural products and medicine [1,2]. They can be used as crucial precursors for constructing novel biomolecules [3,4]. Over the past few decades, many studies have been devoted to these molecular skeletons [5,6,7]. However, most of these conditions are relatively complicated or unfavorable for product purification. Therefore, there is an urgent need to develop a simple, efficient, and straightforward strategy to construct quinoline-2,4-diones. Intramolecular cyclization based on *N*-(2-cyanophenyl)-*N*-methyl-methacrylamide is a very practical idea [8]. According to different mechanisms of the addition to nitriles, it mainly involves two types: (i) nucleophilic addition of metal nucleophiles, such as Grignard reagents [9], lithium reagents [10], and transition metal complexes [11,12] to the cyano group and (ii) radical addition. After continuous research and reports, the cascade/cyclization reaction initiated by radicals has become one of the most effective strategies for constructing the derivatives of quinoline-2,4-diones. Radical addition to highly polar cyano groups are generally not well controlled, due to the generation of unstable imino radicals [13,14]. However, this pathway can become feasible once the unstable imino radical intermediate can be efficiently trapped [15,16]. Usually, such radical reaction conditions still require transition metal catalysts/additives [17,18], such as copper-mediated radical oxidation addition [19,20] and Ag-catalyzed oxidative radical decarboxylation cycloaddition [21].

Visible light-promoted reactions, which typically proceed under mild reaction conditions and offer high efficiency and selectivity, are increasingly gaining enormous attention [22,23]. Light can be considered an ideal reagent for environmentally friendly “green” chemical synthesis; unlike many conventional reagents, light is non-toxic and can be obtained from renewable resources. Generally, visible light-mediated radical addition usually requires the presence of both metal catalysts/additives and photocatalysts [24]. Such reactions require at least one metal catalyst or photocatalyst [25,26,27,28]. Up until now, there is only one report on the cascade cyclization reaction of radical addition under metal-free and photocatalyst-free conditions for the addition to nitriles, and a high temperature (up to 130 °C) was required to generate methyl radicals [29]. Therefore, it is very attractive to develop a transition metal-free and photocatalyst-free photocatalytic method for cascade/cyclization reactions at room temperature, which can simplify the reaction conditions and facilitate subsequent product purification. Based on previous reports [30], we developed a new method to achieve the synthesis of quinoline-2,4-dione derivatives. The general process is that TsSePh generates a sulfonyl-centered radical after light irradiation, which attacks the double bond of *N*-(2-cyanophenyl)-*N*-methyl-methacrylamide to obtain a carbon-centered radical intermediate, and the subsequent cyclization through the intramolecular addition of carbon radicals to nitriles deliver the final quinoline-2,4-diones. When it comes to the synthesis of bioactive molecules, this synthetic strategy can effectively avoid the toxic effects of adding transition metals.

## 2. Results and Discussion

As shown in Table 1, the reaction of *N*-(2-cyanophenyl)-*N*-methylmethacrylamide (**1a**) and TsSePh (**2a**), as the model substrates, was used to optimize the reaction conditions. In the initial reactions, Product **3a** was obtained with a yield of 56% after 2 hours under blue LED irradiation with 10 mmol% Eosin Y as the photocatalysts of CH_3_CN (Entry 1, Table 1). Further optimization showed that with other photocatalysts such as Acid Red, Ru(bpy)_2_Cl_2_·6H_2_O, and 4CZIPN, the corresponding yields for **3a** (Entries 2–4) decreased respectively. As a contrast, when the reaction was conducted without the participation of photocatalysts, the yield of **3a** was about 50% (Entry 5), indicating that the photocatalysts were not integral to this process. When DMSO, DCM, and THF were used as reaction solvents, the corresponding yields for **3a** were not as efficient as those with CH_3_CN (Entries 6–8). Considering the necessity of water to produce carbonyl groups in **3a**, mixed solvents were tested. To our delight, mixed solvents with water such as CH_3_CN/H_2_O and DMSO/H_2_O effectively increased the yield of the reaction, and CH_3_CN/H_2_O (2/1) was found to be the best, wherein the yield of **3a** unexpectedly reached up to 80%, which indicates that the addition of water can effectively increase the reaction yield (Entries 9–11). If the reaction was only carried out for one hour, the results showed that it had obviously not completed with a low yield of **3a** obtained (55%) (Entry 12).

With optimal conditions in hand, we subsequently investigated the substrate scope of anthranilonitrile derivatives **1** under the conditions of this photocatalytic system for cascade sulfonylation/cyclization reactions. As shown in Table 2, moderate yields (**3b**–**3d**) were obtained when substrate **1** contained an electron-withdrawing group on the benzene ring ortho to the cyano group. The meta-position of the cyano group on the benzene ring showed good tolerance, regardless of the introduction of electron-donating groups or electron-withdrawing groups, and the corresponding products **3e**–**3i** were also obtained with moderate to good yields (61–79%). Similarly, more diverse electron-donating and electron-withdrawing groups at the related para-positions produced the corresponding products **3j**–**3q** in satisfactory yields (61–81%). Moreover, for substrate **1r** (both the meta-position and the para-position contain -OMe groups) and substrate **1s** (a benzyl group attached to the N atom), the corresponding products for **3r** and **3s** were obtained as 58% and 55%, respectively. These results indicate that the mild reaction conditions are compatible with a wide range of functional groups.

In view of the previous report on the visible light-induced tandem reaction of allene and selensulfonate24, the process is mainly due to photoinduced radical addition through energy transfer. Therefore, we tried to introduce different derivatives of TsSePh into our reaction system as the radical sources. As shown in Table 3, the corresponding target molecules **3t**–**3y** were obtained in good yields (51–81%), indicating a wide functional group tolerance. Specifically, for the *p*-toluenesulfonyl part of **2**, when methyl was replaced by -H, -OMe or halogen atom (-F, -Cl, -Br), the reaction proceeded smoothly in good yields (74–81%). Notably, derivatives of **2** bearing cyclopropyl or naphthalene were also compatible with this cascade reaction, affording the desired products **3w** and **3x** with 71% and 75% yields, respectively.

In addition, we also expanded the reaction from **1a** to **3a** on the gram level, and the reaction time was extended to 5 h, which still maintained a high yield of **3a** (76%) (Figure 1a). By using **3a** as a starting material, the target product **4** could be obtained after sodium borohydride reduction (Figure 1b) [31]. By using the Cu/NFSI system [32], efficient conversion of **3a** from the tertiary amine to the secondary amine **5** was achieved (Figure 1c). Both **4** with -OH and **5** with -NH can be used as very useful intermediates, as then other different functionalized molecules can be synthesized from **4** and **5**.

In the radical-trapping experiment, by adding the radical scavenger 2,2,6,6-tetramethylpiperidinyl-1-oxyl (TEMPO) to the cascade reaction, the target product **3a** was only detected in trace amounts, and the adduct **6** was detected by NMR analysis (Figure 2a). Likewise, this cascade was also inhibited when two other radical inhibitors 3,5-di-tert-butyl-4-hydroxytoluene (BHT) or 1,1-diphenylethylene were used, with the adducts **7** and **8** detected by NMR analysis (Figure 2b,c). All these results indicate that this photocatalytic reaction had undergone a radical pathway, and sulfonyl-centered radical was involved during the transformation.

Based on the above results and previous reports [11], a plausible mechanism was proposed. As described in Figure 3, Ts and SePh radicals were generated initially under visible irradiation. Subsequently, the addition of sulfonyl-centered radical to the double bond of **1a** yielded the alkyl radical intermediate **A**, which then underwent intramolecular addition to polar cyano groups and gave the imino radical intermediate **B**. The intermediate imine **C** was obtained after the H-absorption of radical **B**. Finally, imine **C** was hydrolyzed by H_2_O to yield the target product **3a**. Therefore, this mechanism shows that mixed solvent with water can significantly increase the yield.

## 3. Materials and Methods

### 3.1. Materials and Instruments

All reagents were purchased from commercial sources and used without further purification. ^1^H-NMR, ^13^C-NMR, and ^19^F-NMR spectra were recorded on a Bruker Ascend™ 500 spectrometer in deuterated solvents containing TMS as an internal reference standard. All high-resolution mass spectra (HRMS) were measured on a mass spectrometer by using electrospray ionization orthogonal acceleration time-of-flight (ESI-OA-TOF), and the purity of all samples used for HRMS (>95%) was confirmed by ^1^H-NMR and ^13^C-NMR spectroscopic analysis. Melting points were measured on a melting point apparatus equipped with a thermometer and were uncorrected.

In these photochemical experiments, we used a 30-W blue LED (455–460 nm), with the reaction bottle 2 cm from the light source. All the reactions were monitored by thin-layer chromatography (TLC) using GF254 silica gel-coated TLC plates. Purification by flash column chromatography was performed over SiO_2_ (silica gel 200–300 mesh).

### 3.2. The General Procedure for the Synthesis of ***3***

In a reaction tube, acrylamides **1** (0.5 mmol) and PhSeSO_2_R **2** (1.2 equiv, 0.6 mmol) were mixed in CH_3_CN/H_2_O (2:1, 3 mL) and irradiated for 2 h until complete consumption of starting material, as monitored by TLC analysis. After the completion of the reaction, the mixture was quenched by NaHCO_3_ (sat. aq. 10 mL) and extracted with CH_2_Cl_2_ (3 × 10 mL). Then the organic solvent was concentrated in vacuo. The residue was purified by flash column chromatography with ethyl acetate and petroleum ether as eluent to product **3**.

***1,3-dimethyl-3-(tosylmethyl)quinoline-2,4(1H,3H)-dione* (3a)**. The product was purified by column chromatography on silica gel (petroleum ether/ethyl acetate = 5:1, **R***_f_* = 0.25), white solid (95 mg, 89%): mp: 161–162 °C. **^1^H-NMR** (500 MHz, CDCl_3_) *δ* 8.09 (dd, *J* = 7.7, 1.5 Hz, 1H), 7.84–7.61 (m, 3H), 7.41–7.18 (m, 4H), 4.22 (s, 2H), 3.54 (s, 3H), 2.43 (s, 3H), 1.41 (s, 3H). **^13^C-NMR** (126 MHz, CDCl_3_) *δ* 194.28, 171.69, 144.52, 143.15, 138.64, 136.50, 129.63, 128.66, 127.83, 123.29, 119.23, 115.16, 62.40, 55.25, 30.14, 25.91, 21.65. **HRMS** (ESI) calculated for C_19_H_20_NO_4_S [M+H]^+^: 358.1108, found: 358.1103.

***5-fluoro-1,3-dimethyl-3-(tosylmethyl)quinoline-2,4(1H,3H)-dione* (3b)**. The product was purified by column chromatography on silica gel (petroleum ether/ethyl acetate = 5:1, **R***_f_* = 0.23), white solid (52 mg, 49%): mp: 206–207 °C. **^1^H-NMR** (500 MHz, CDCl_3_) *δ* 7.74 (d, *J* = 8.3 Hz, 2H), 7.61 (td, *J* = 8.4, 5.8 Hz, 1H), 7.33 (d, *J* = 8.1 Hz, 2H), 7.05 (d, *J* = 8.5 Hz, 1H), 6.91 (dd, *J* = 9.9, 8.7 Hz, 1H), 4.19 (d, *J* = 17.0 Hz, 2H), 3.55 (s, 3H), 2.44 (s, 3H), 1.43 (s, 3H). **^13^C-NMR** (126 MHz, CDCl_3_) *δ* 191.50, 171.31, 162.74 (d, *J* = 267.2 Hz), 144.53, 144.29 (d, *J* = 2.8 Hz), 138.79, 136.69 (d, *J* = 11.9 Hz), 129.68, 127.83, 111.46 (d, *J* = 21.3 Hz), 111.02 (d, *J* = 3.5 Hz), 109.09 (d, *J* = 9.0 Hz), 61.93, 56.31, 30.97, 25.43, 21.65. **^19^F-NMR** (471 MHz, CDCl_3_) *δ* -109.25. **HRMS** (ESI) calculated for C_19_H_19_FNO_4_S [M+H]^+^: 376.1013, found: 376.1009.

***5-chloro-1,3-dimethyl-3-(tosylmethyl)quinoline-2,4(1H,3H)-dione* (3c)**. The product was purified by column chromatography on silica gel (petroleum ether/ethyl acetate = 5:1, **R***_f_* = 0.30), white solid (62 mg, 53%): mp: 182–183 °C. **^1^H-NMR** (500 MHz, CDCl_3_) *δ* 7.79–7.67 (m, 2H), 7.46–7.41 (m, 1H), 7.32–7.25 (m, 2H), 7.20–7.17 (m, 1H), 7.12 (t, *J* = 9.1 Hz, 1H), 4.11 (d, *J* = 7.7 Hz, 2H), 3.49 (s, 3H), 2.37 (s, 3H), 1.34 (s, 3H). **^13^C-NMR** (126 MHz, CDCl_3_) *δ* 190.97, 169.97, 143.88, 143.43, 138.00, 135.40, 133.79, 128.61, 126.86, 125.75, 116.13, 113.02, 60.77, 55.72, 30.06, 23.67, 20.62. **HRMS** (ESI) calculated for C_19_H_19_ClNO_4_S [M+H]^+^: 392.0718, found: 392.0710.

***5-bromo-1,3-dimethyl-3-(tosylmethyl)quinoline-2,4(1H,3H)-dione* (3d)**. The product was purified by column chromatography on silica gel (petroleum ether/ethyl acetate = 5:1, **R***_f_* = 0.30), white solid (67 mg, 51%): mp: 140–141 °C. **^1^H-NMR** (500 MHz, CDCl_3_) *δ* 7.76 (d, *J* = 8.2 Hz, 2H), 7.48–7.40 (m, 2H), 7.34 (d, *J* = 8.1 Hz, 2H), 7.25–7.22 (m, 1H), 4.19 (d, *J* = 4.7 Hz, 2H), 3.55 (s, 3H), 2.44 (s, 3H), 1.40 (s, 3H). **^13^C-NMR** (126 MHz, CDCl_3_) *δ* 192.19, 170.87, 145.01, 144.47, 139.04, 135.07, 130.43, 129.84, 129.65, 127.88, 123.89, 114.84, 61.80, 56.52, 31.03, 24.57, 21.66. **HRMS** (ESI) calculated for C_19_H_19_BrNO_4_S [M+H]^+^: 436.0213, found: 436.0200.

***5-fluoro-1,3-dimethyl-3-(tosylmethyl)quinoline-2,4(1H**,3H)-dione* (3e)**. The product was purified by column chromatography on silica gel (petroleum ether/ethyl acetate = 5:1, **R***_f_* = 0.23), white solid (87 mg, 77%): mp: 135–136 °C. **^1^H-NMR** (500 MHz, CDCl_3_) *δ* 7.69 (dd, *J* = 8.0, 3.1 Hz, 1H), 7.62 (d, *J* = 8.2 Hz, 2H), 7.31 (ddd, *J* = 9.2, 7.5, 3.1 Hz, 1H), 7.25 (d, *J* = 8.1 Hz, 2H), 7.15 (dd, *J* = 9.1, 4.0 Hz, 1H), 4.14 (d, *J* = 5.7 Hz, 2H), 3.46 (s, 3H), 2.36 (s, 3H), 1.34 (s, 3H). **^13^C-NMR** (126 MHz, CDCl_3_) *δ* 193.60 (d, *J* = 1.6 Hz), 171.27, 158.58 (d, *J* = 245.4 Hz), 144.65, 139.61 (d, *J* = 1.9 Hz), 138.50, 129.69, 127.80, 123.49 (d, *J* = 23.3 Hz), 120.40 (d, *J* = 6.4 Hz), 117.06 (d, *J* = 7.2 Hz), 114.28 (d, *J* = 23.4 Hz), 62.54, 55.05, 30.41, 25.78, 21.65. **^19^F-NMR** (471 MHz, CDCl_3_) *δ* −119.37. **HRMS** (ESI) calculated for C_19_H_19_FNO_4_S [M+H]^+^: 376.1013, found: 376.1009.

***6-chloro-1,3-dimethyl-3-(tosylmethyl)quinoline-2,4(1H,3H)-dione* (3f)**. The product was purified by column chromatography on silica gel (petroleum ether/ethyl acetate = 5:1, **R***_f_* = 0.23), white solid (93 mg, 79%): mp: 139–140 °C. **^1^H-NMR** (500 MHz, CDCl_3_) *δ* 7.76 (d, *J* = 8.2 Hz, 2H), 7.49 (d, *J* = 7.8 Hz, 1H), 7.43 (d, *J* = 8.2 Hz, 1H), 7.34 (d, *J* = 8.1 Hz, 2H), 7.23 (d, *J* = 8.3 Hz, 1H), 4.19 (d, *J* = 4.8 Hz, 2H), 3.56 (s, 3H), 2.44 (s, 3H), 1.41 (s, 3H). **^13^C-NMR** (126 MHz, CDCl_3_) *δ* 193.25, 171.76, 144.67, 142.85, 138.42, 130.05, 129.70, 127.77, 126.41, 123.59, 117.60, 115.53, 62.42, 55.21, 30.29, 25.82, 21.66. **HRMS** (ESI) calculated for C_19_H_19_ClNO_4_S [M+H]^+^: 392.0718, found: 392.0710.

***5-bromo-1,3-dimethyl-3-(tosylmethyl)quinoline-2,4(1H,3H)-dione* (3g)**. The product was purified by column chromatography on silica gel (petroleum ether/ethyl acetate = 5:1, **R***_f_* = 0.21), white solid (80 mg, 61%): mp: 148–149 °C. **^1^H-NMR** (500 MHz, CDCl_3_) *δ* 8.00 (dd, *J* = 43.4, 8.3 Hz, 1H), 7.70 (d, *J* = 8.2 Hz, 2H), 7.44–7.20 (m, 4H), 4.20 (s, 2H), 3.53 (s, 3H), 2.44 (s, 3H), 1.41 (s, 3H). **^13^C-NMR** (126 MHz, CDCl_3_) *δ* 193.22, 171.76, 144.63, 142.85, 138.49, 130.09, 129.69, 127.80, 126.55, 123.59, 118.42, 115.48, 62.51, 55.23, 30.28, 25.84, 21.66. **HRMS** (ESI) calculated for C_19_H_19_BrNO_4_S [M+H]^+^: 436.0213, found: 436.0200.

***1,3,6-trimethyl-3-(tosylmethyl)quinoline-2,4(1H,3H)-dione* (3h)**. The product was purified by column chromatography on silica gel (petroleum ether/ethyl acetate = 5:1, **R***_f_* = 0.23), white solid (80 mg, 72%): mp: 174–175 °C. **^1^H-NMR** (500 MHz, CDCl_3_) *δ* 7.89 (d, *J* = 1.7 Hz, 1H), 7.71 (d, *J* = 8.3 Hz, 2H), 7.47 (dd, *J* = 8.4, 2.1 Hz, 1H), 7.31 (d, *J* = 8.2 Hz, 2H), 7.13 (d, *J* = 8.5 Hz, 1H), 4.21 (s, 2H), 3.51 (s, 3H), 2.43 (s, 3H), 2.38 (s, 3H), 1.40 (s, 3H). **^13^C-NMR** (126 MHz, CDCl_3_) *δ* 194.48, 171.49, 144.45, 141.00, 138.68, 137.31, 133.01, 129.60, 128.53, 127.83, 119.02, 115.17, 62.42, 55.10, 30.09, 25.96, 21.65, 20.35. **HRMS** (ESI) calculated for C_20_H_22_NO_4_S [M+H]^+^: 372.1264, found: 372.1261.

***6-methoxy-1,3-dimethyl-3-(tosylmethyl)quinoline-2,4(1H,3H)-dione* (3i)**. The product was purified by column chromatography on silica gel (petroleum ether/ethyl acetate = 5:1, **R***_f_* = 0.24), yellow solid (82 mg, 71%): mp: 87–88 °C. **^1^H-NMR** (500 MHz, CDCl_3_) *δ* 7.72 (d, *J* = 8.3 Hz, 2H), 7.57 (d, *J* = 3.1 Hz, 1H), 7.32 (d, *J* = 8.0 Hz, 2H), 7.28–7.24 (m, 1H), 7.17 (d, *J* = 9.0 Hz, 1H), 4.21 (d, *J* = 2.5 Hz, 2H), 3.87 (s, 3H), 3.52 (s, 3H), 2.44 (s, 3H), 1.41 (s, 3H). **^13^C-NMR** (126 MHz, CDCl_3_) *δ* 194.39, 171.18, 155.59, 144.48, 138.63, 137.34, 129.62, 127.85, 124.52, 119.86, 116.70, 110.19, 62.50, 55.83, 54.94, 30.21, 26.05, 21.65. **HRMS** (ESI) calculated for C_20_H_22_NO_5_S [M+H]^+^: 383.1213, found: 383.1234.

***7-fluoro-1,3-dimethyl-3-(tosylmethyl)quinoline-2,4(1H,3H)-dione* (3j)**. The product was purified by column chromatography on silica gel (petroleum ether/ethyl acetate = 5:1, **R***_f_* = 0.21), yellow solid (86 mg, 76%): mp: 114–115 °C. **^1^H-NMR** (500 MHz, CDCl_3_) *δ* 8.24–8.08 (m, 1H), 7.70 (d, *J* = 8.3 Hz, 2H), 7.32 (d, *J* = 8.2 Hz, 2H), 6.92 (d, *J* = 9.2 Hz, 2H), 4.20 (s, 2H), 3.52 (s, 3H), 2.44 (s, 3H), 1.41 (s, 3H). **^13^C-NMR** (126 MHz, CDCl_3_) *δ* 192.79, 171.87, 167.81 (d, *J* = 256.5 Hz), 145.48 (d, *J* = 11.8 Hz), 144.62, 138.50, 131.67 (d, *J* = 11.4 Hz), 129.68, 127.79, 115.88 (d, *J* = 2.5 Hz), 110.75 (d, *J* = 22.4 Hz), 102.79 (d, *J* = 27.6 Hz), 62.44, 55.07, 30.32, 25.91, 21.66. **^19^F-NMR** (471 MHz, CDCl_3_) *δ* −98.47. **HRMS** (ESI) calculated for C_19_H_19_FNO_4_S [M+H]^+^: 376.1013, found: 376.1008.

***7-chloro-1,3-dimethyl-3-(tosylmethyl)quinoline-2,4(1H,3H)-dione* (3k)**. The product was purified by column chromatography on silica gel (petroleum ether/ethyl acetate = 5:1, **R***_f_* = 0.23), yellow solid (84 mg, 71%): mp: 132–133 °C. **^1^H-NMR** (500 MHz, CDCl_3_) *δ* 8.03 (d, *J* = 2.5 Hz, 1H), 7.69 (d, *J* = 8.2 Hz, 2H), 7.60 (dd, *J* = 8.8, 2.6 Hz, 1H), 7.33 (d, *J* = 8.3 Hz, 2H), 7.20 (d, *J* = 8.9 Hz, 1H), 4.21 (d, *J* = 2.3 Hz, 2H), 3.52 (s, 3H), 2.44 (s, 3H), 1.41 (s, 3H). **^13^C-NMR** (126 MHz, CDCl_3_) *δ* 193.36, 171.35, 144.68, 141.69, 138.45, 136.11, 129.70, 129.06, 127.96, 127.77, 120.19, 116.96, 62.51, 55.21, 30.32, 25.73, 21.66. **HRMS** (ESI) calculated for C_19_H_19_ClNO_4_S [M+H]^+^: 392.0718, found: 392.0712.

***7-bromo-1,3-dimethyl-3-(tosylmethyl)quinoline-2,4(1H,3H)-dione* (3l)**. The product was purified by column chromatography on silica gel (petroleum ether/ethyl acetate = 5:1, **R***_f_* = 0.23), white solid (95 mg, 81%): mp: 174–175 °C. **^1^H-NMR** (500 MHz, CDCl_3_) *δ* 7.95 (d, *J* = 8.3 Hz, 1H), 7.69 (d, *J* = 8.2 Hz, 2H), 7.41 (d, *J* = 0.8 Hz, 1H), 7.32 (d, *J* = 8.1 Hz, 3H), 4.20 (s, 2H), 3.53 (s, 3H), 2.43 (s, 3H), 1.40 (s, 3H). **^13^C-NMR** (126 MHz, CDCl_3_) *δ* 193.42, 171.69, 144.63, 143.97, 138.48, 131.60, 129.97, 129.68, 127.79, 126.53, 118.43, 117.97, 62.49, 55.23, 30.28, 25.80, 21.66. **HRMS** (ESI) calculated for C_19_H_19_BrNO_4_S [M+H]^+^: 392.0718, found: 392.0710.

***1,3,7-trimethyl-3-(tosylmethyl)quinoline-2,4(1H,3H)-dione* (3m)**. The product was purified by column chromatography on silica gel (petroleum ether/ethyl acetate = 5:1, **R***_f_* = 0.22), white solid (83 mg, 74%): mp: 137–138 °C. **^1^H-NMR** (500 MHz, CDCl_3_) *δ* 8.03–7.95 (m, 1H), 7.70 (d, *J* = 8.3 Hz, 2H), 7.31 (d, *J* = 8.2 Hz, 2H), 7.03 (s, 2H), 4.20 (s, 2H), 3.53 (s, 3H), 2.47 (s, 3H), 2.43 (s, 3H), 1.40 (s, 3H). **^13^C-NMR** (126 MHz, CDCl_3_) *δ* 193.80, 171.94, 147.98, 144.44, 143.21, 138.68, 129.60, 128.71, 127.84, 124.34, 117.11, 115.60, 62.43, 54.98, 30.07, 26.08, 22.49, 21.65. **HRMS** (ESI) calculated for C_20_H_22_NO_4_S [M+H]^+^: 372.1264, found: 372.1261.

***7-methoxy-1,3-dimethyl-3-(tosylmethyl)quinoline-2,4(1H,3H)-dione* (3n)**. The product was purified by column chromatography on silica gel (petroleum ether/ethyl acetate = 5:1, **R***_f_* = 0.23), yellow solid (83 mg, 71%): mp: 192–193 °C. **^1^H-NMR** (500 MHz, CDCl_3_) *δ* 7.72 (d, *J* = 8.3 Hz, 2H), 7.57 (d, *J* = 3.1 Hz, 1H), 7.32 (d, *J* = 8.0 Hz, 2H), 7.29–7.24 (m, 1H), 7.17 (d, *J* = 9.0 Hz, 1H), 4.21 (d, *J* = 2.4 Hz, 2H), 3.87 (s, 3H), 3.52 (s, 3H), 2.43 (s, 3H), 1.41 (s, 3H). **^13^C-NMR** (126 MHz, CDCl_3_) *δ* 194.39, 171.17, 155.58, 144.48, 138.62, 137.33, 129.62, 127.85, 124.51, 119.85, 116.72, 110.20, 62.49, 55.83, 54.94, 30.20, 26.04, 21.65. **HRMS** (ESI) calculated for C_20_H_22_NO_4_S [M+H]^+^: 388.1213, found: 383.1235.

***1,3-dimethyl-7-phenyl-3-(tosylmethyl)quinoline-2,4(1H,3H)-dione* (3o)**. The product was purified by column chromatography on silica gel (petroleum ether/ethyl acetate = 5:1, **R***_f_* = 0.23), white solid (83 mg, 64%): mp: 100–101 °C. **^1^H-NMR** (500 MHz, CDCl_3_) *δ* 8.16 (d, *J* = 8.0 Hz, 1H), 7.72 (d, *J* = 8.2 Hz, 2H), 7.64 (d, *J* = 7.3 Hz, 2H), 7.54–7.38 (m, 5H), 7.32 (d, *J* = 8.1 Hz, 2H), 4.23 (s, 2H), 3.62 (s, 3H), 2.44 (s, 3H), 1.45 (s, 3H). **^13^C-NMR** (126 MHz, CDCl_3_) *δ* 193.94, 171.94, 149.61, 144.53, 143.54, 139.81, 138.63, 129.66, 129.28, 129.13, 128.92, 127.85, 127.44, 126.46, 122.34, 118.00, 113.87, 62.50, 55.16, 30.23, 26.04, 21.67. **HRMS** (ESI) calculated for C_25_H_24_NO_4_S [M+H]^+^: 434.1421, found: 434.1410.

***7-(4-acetylphenyl)-1,3-dimethyl-3-(tosylmethyl)quinoline-2,4(1H,3H)-dione* (3p)**. The product was purified by column chromatography on silica gel (petroleum ether/ethyl acetate = 5:1, **R***_f_* = 0.23), yellow solid (101 mg, 88%): mp: 192–193 °C. **^1^H-NMR** (500 MHz, CDCl_3_) *δ* 8.19 (d, *J* = 8.0 Hz, 1H), 8.09 (d, *J* = 8.3 Hz, 2H), 7.73 (t, *J* = 8.0 Hz, 4H), 7.48–7.41 (m, 2H), 7.33 (d, *J* = 8.1 Hz, 2H), 4.24 (s, 2H), 3.63 (s, 3H), 2.66 (s, 3H), 2.44 (s, 3H), 1.45 (s, 3H). **^13^C-NMR** (126 MHz, CDCl_3_) *δ* 197.54, 193.88, 171.85, 148.08, 144.59, 144.15, 143.62, 138.60, 137.06, 129.68, 129.42, 129.11, 127.81, 127.68, 122.35, 118.56, 114.03, 62.55, 55.22, 30.27, 26.77, 25.94, 21.67. **HRMS** (ESI) calculated for C_27_H_26_NO_5_S [M+H]^+^: 476.1526, found: 476.1520.

***7-cyclopropyl-1,3-dimethyl-3-(tosylmethyl)quinoline-2,4(1H,3H)-dione* (3q)**. The product was purified by column chromatography on silica gel (petroleum ether/ethyl acetate = 5:1, **R***_f_* = 0.22), white solid (73 mg, 61%): mp: 151–152 °C. **^1^H-NMR** (500 MHz, CDCl_3_) *δ* 7.97 (d, *J* = 8.1 Hz, 1H), 7.70 (d, *J* = 8.3 Hz, 2H), 7.29 (t, *J* = 10.3 Hz, 2H), 6.93 (d, *J* = 0.7 Hz, 1H), 6.82 (dd, *J* = 8.1, 1.1 Hz, 1H), 4.19 (s, 2H), 3.54 (s, 3H), 2.43 (s, 3H), 2.04–1.95 (m, 1H), 1.40 (s, 3H), 1.19–1.10 (m, 2H), 0.91–0.82 (m, 2H). **^13^C-NMR** (126 MHz, CDCl_3_) *δ* 193.58, 172.02, 154.61, 144.43, 143.24, 138.67, 129.59, 128.82, 127.84, 119.83, 116.96, 112.53, 62.42, 54.90, 30.05, 26.12, 21.65, 16.62, 10.77, 10.75. **HRMS** (ESI) calculated for C_22_H_24_NO_4_S [M+H]^+^: 398.1421, found: 398.1420.

***1-benzyl-3-methyl-3-(tosylmethyl)quinoline-2,4(1H,3H)-dione* (3r)**. The product was purified by column chromatography on silica gel (petroleum ether/ethyl acetate = 5:1, **R***_f_* = 0.21), white solid (72 mg, 55%): mp: 166–167 °C. **^1^H-NMR** (500 MHz, CDCl_3_) *δ* 8.11 (dd, *J* = 7.8, 1.6 Hz, 1H), 7.74 (d, *J* = 8.3 Hz, 2H), 7.49 (s, 1H), 7.33 (ddd, *J* = 18.7, 10.8, 7.7 Hz, 6H), 7.25 (d, *J* = 4.6 Hz, 1H), 7.16 (d, *J* = 7.5 Hz, 1H), 7.08 (d, *J* = 8.4 Hz, 1H), 5.53–5.23 (m, 2H), 4.30 (s, 2H), 2.42 (s, 3H), 1.50 (s, 3H). **^13^C-NMR** (126 MHz, CDCl_3_) *δ* 194.15, 172.30, 144.47, 142.28, 138.94, 136.33, 135.88, 129.65, 128.99, 128.81, 127.84, 127.41, 126.38, 123.37, 119.43, 116.15, 62.28, 55.78, 46.44, 25.92, 21.66. **HRMS** (ESI) calculated for C_25_H_24_NO_4_S [M+H]^+^: 434.1421, found: 434.1414.

***6,7-dimethoxy-1,3-dimethyl-3-(tosylmethyl)quinoline-2,4(1H,3H)-dione* (3s)**. The product was purified by column chromatography on silica gel (petroleum ether/ethyl acetate = 5:1, **R***_f_* = 0.23), yellow liquid (73 mg, 58%): mp: 180–181 °C. **^1^H-NMR** (500 MHz, CDCl_3_) *δ* 7.71 (d, *J* = 8.3 Hz, 2H), 7.53 (s, 1H), 7.31 (d, *J* = 8.1 Hz, 2H), 6.67 (s, 1H), 4.19 (d, *J* = 6.5 Hz, 2H), 4.02 (s, 3H), 3.94 (s, 3H), 3.55 (s, 3H), 2.43 (s, 3H), 1.41 (s, 3H). **^13^C-NMR** (126 MHz, CDCl_3_) *δ* 192.79, 172.08, 156.01, 145.42, 144.45, 139.45, 138.57, 129.61, 127.82, 111.92, 109.19, 98.39, 62.52, 56.41, 56.25, 54.40, 30.17, 26.49, 21.64. **HRMS** (ESI) calculated for C_21_H_24_NO_6_S [M+H]^+^: 418.1319, found: 418.1315.

***1,3-dimethyl-3-((phenylsulfonyl)methyl)quinoline-2,4(1H,3H)-dione* (3t)**. The product was purified by column chromatography on silica gel (petroleum ether/ethyl acetate = 5:1, **R***_f_* = 0.23), white solid (77 mg, 75%): mp: 118–119 °C. **^1^H-NMR** (500 MHz, CDCl_3_) *δ* 8.10 (dd, *J* = 7.7, 1.4 Hz, 1H), 7.85 (d, *J* = 7.4 Hz, 2H), 7.71–7.51 (m, 4H), 7.24 (dt, *J* = 9.8, 8.9 Hz, 2H), 4.24 (s, 2H), 3.55 (s, 3H), 1.43 (s, 3H). **^13^C-NMR** (126 MHz, CDCl_3_) *δ* 194.26, 171.67, 143.15, 141.55, 136.53, 133.58, 129.03, 128.70, 127.81, 123.34, 119.23, 115.17, 62.21, 55.32, 30.16, 25.91. **HRMS** (ESI) calculated for C_18_H_18_NO_4_S [M+H]^+^: 344.0951, found: 344.0947.

***3-(((4-bromophenyl)sulfonyl)methyl)-1,3-dimethylquinoline-2,4(1H,3H)-dione* (3u)**. The product was purified by column chromatography on silica gel (petroleum ether/ethyl acetate = 5:1, **R***_f_* = 0.23), white solid (102 mg, 81%): mp: 174–175 °C. **^1^H-NMR** (500 MHz, CDCl_3_) *δ* 8.09 (dd, *J* = 7.7, 1.5 Hz, 1H), 8.00 (d, *J* = 8.2 Hz, 2H), 7.82 (d, *J* = 8.3 Hz, 2H), 7.70 (s, 1H), 7.26 (dd, *J* = 8.0, 2.6 Hz, 2H), 4.27 (s, 2H), 3.56 (s, 3H), 1.43 (s, 3H). **^13^C-NMR** (126 MHz, CDCl_3_) *δ* 194.26, 171.63, 145.11, 143.05, 136.68, 128.69, 128.54, 126.18, 126.15, 123.49, 119.13, 115.24, 61.82, 55.78, 30.20, 25.75. **HRMS** (ESI) calculated for C_18_H_17_BrNO_4_S [M+H]^+^: 422.0056, found: 422.0051.

***3-(((4-chlorophenyl)sulfonyl)methyl)-1,3-dimethylquinoline-2,4(1H,3H)-dione* (3v)**. The product was purified by column chromatography on silica gel (petroleum ether/ethyl acetate = 5:1, **R***_f_* = 0.23), white solid (86 mg, 76%): mp: 180–181 °C. **^1^H-NMR** (500 MHz, CDCl_3_) *δ* 8.08 (d, *J* = 7.6 Hz, 1H), 7.78 (d, *J* = 8.5 Hz, 2H), 7.68 (s, 1H), 7.50 (d, *J* = 8.5 Hz, 2H), 7.32–7.18 (m, 2H), 4.24 (s, 2H), 3.54 (s, 3H), 1.42 (s, 3H). **^13^C-NMR** (126 MHz, CDCl_3_) *δ* 194.26, 171.63, 143.07, 140.25, 140.13, 136.64, 129.40, 129.31, 128.64, 123.41, 119.13, 115.24, 62.10, 55.60, 30.18, 25.82. **HRMS** (ESI) calculated for C_18_H_16_ClNNaO_4_S [M+H]^+^: 400.0381, found: 400.0411.

***3-((cyclopropylsulfonyl)methyl)-1,3-dimethylquinoline-2,4(1H,3H)-dione* (3w)**. The product was purified by column chromatography on silica gel (petroleum ether/ethyl acetate = 5:1, **R***_f_* = 0.25), yellow liquid (66 mg, 71%): mp: 153–155 °C. **^1^H-NMR** (500 MHz, CDCl_3_) *δ* 8.08 (dd, *J* = 8.1, 1.7 Hz, 1H), 7.72–7.62 (m, 1H), 7.28–7.17 (m, 2H), 4.21 (d, *J* = 1.3 Hz, 2H), 3.53 (s, 3H), 2.70–2.58 (m, 1H), 1.46 (s, 3H), 1.24–1.15 (m, 2H), 1.00 (dd, *J* = 8.0, 2.0 Hz, 2H). **^13^C-NMR** (126 MHz, CDCl_3_) *δ* 194.82, 172.13, 143.04, 136.48, 128.63, 123.33, 119.14, 115.14, 59.87, 55.72, 33.32, 30.14, 25.64, 5.13, 5.02. **HRMS** (ESI) calculated for C_15_H_18_NO_4_S [M+H]^+^: 308.0951, found: 308.0947.

***1,3-dimethyl-3-((naphthalen-2-ylsulfonyl)methyl)quinoline-2,4(1H,3H)-dione* (3x)**. The product was purified by column chromatography on silica gel (petroleum ether/ethyl acetate = 5:1, **R***_f_* = 0.24), white solid (89 mg, 75%): mp: 135–136 °C. **^1^H-NMR** (500 MHz, CDCl_3_) *δ* 8.35 (s, 1H), 8.08 (d, *J* = 7.3 Hz, 1H), 7.98 (d, *J* = 8.7 Hz, 1H), 7.91 (t, *J* = 7.3 Hz, 2H), 7.85 (dd, *J* = 8.6, 1.4 Hz, 1H), 7.72–7.53 (m, 3H), 7.21 (dd, *J* = 12.2, 5.4 Hz, 2H), 4.30 (s, 2H), 3.51 (s, 3H), 1.43 (s, 3H). **^13^C-NMR** (126 MHz, CDCl_3_) *δ* 194.19, 171.61, 143.14, 138.20, 136.58, 135.30, 131.99, 129.66, 129.53, 129.42, 129.21, 128.67, 127.98, 127.53, 123.34, 122.63, 119.25, 115.20, 62.30, 55.17, 30.14, 26.09. **HRMS** (ESI) calculated for C_22_H_20_NO_4_S [M+H]^+^: 394.1180, found: 394.1094.

***3-(((4-(tert-butyl)phenyl)sulfonyl)methyl)-1,3-dimethylquinoline-2,4(1H,3H)-dione* (3y)**. The product was purified by column chromatography on silica gel (petroleum ether/ethyl acetate = 5:1, **R***_f_* = 0.25), white liquid (61 mg, 51%): mp: 95–96 °C. **^1^H-NMR** (500 MHz, CDCl_3_) *δ* 8.11 (dd, *J* = 7.7, 1.6 Hz, 1H), 7.80–7.75 (m, 2H), 7.76–7.68 (m, 1H), 7.56–7.50 (m, 2H), 7.27–7.21 (m, 2H), 4.23 (s, 2H), 3.55 (s, 3H), 1.42 (s, 3H), 1.35 (s, 9H). **^13^C-NMR** (126 MHz, CDCl_3_) *δ* 194.31, 171.73, 157.40, 143.17, 138.50, 136.47, 128.70, 127.69, 126.05, 123.29, 119.26, 115.15, 62.36, 55.26, 35.24, 31.09, 30.16, 25.89. **HRMS** (ESI) calculated for C_22_H_26_NO_4_S [M+H]^+^: 400.1577, found: 400.1572.

***3-(((4-methoxyphenyl)sulfonyl)methyl)-1,3-dimethylquinoline-2,4(1H,3H)-dione* (3z)**. The product was purified by column chromatography on silica gel (petroleum ether/ethyl acetate = 5:1, **R***_f_* = 0.25), white solid (83 mg, 74%): mp: 145–146 °C. **^1^H-NMR** (500 MHz, CDCl_3_) *δ* 7.99 (dd, *J* = 7.7, 1.4 Hz, 1H), 7.66 (d, *J* = 8.9 Hz, 2H), 7.58 (s, 1H), 7.18–7.09 (m, 2H), 6.89 (d, *J* = 8.9 Hz, 2H), 4.13 (s, 2H), 3.77 (s, 3H), 3.44 (s, 3H), 1.33 (s, 3H). **^13^C-NMR** (126 MHz, CDCl_3_) *δ* 194.33, 171.73, 163.63, 143.16, 136.49, 133.15, 130.05, 128.66, 123.28, 119.25, 115.15, 114.16, 62.63, 55.67, 55.26, 30.14, 25.92. **HRMS** (ESI) calculated for C_19_H_19_NO_5_S [M+H]^+^: 374.1057, found: 374.1052.

***3-(((4-fluorophenyl)sulfonyl)methyl)-1,3-dimethylquinoline-2,4(1H,3H)-dione* (3aa)**. The product was purified by column chromatography on silica gel (petroleum ether/ethyl acetate = 5:1, **R***_f_* = 0.21), white solid (83 mg, 76%): mp: 147-148 °C. **^1^H-NMR** (500 MHz, CDCl_3_) *δ* 8.09 (dd, *J* = 7.7, 1.5 Hz, 1H), 7.92–7.79 (m, 2H), 7.72–7.64 (m, 1H), 7.29–7.16 (m, 4H), 4.25 (s, 2H), 3.55 (s, 3H), 1.42 (s, 3H). **^13^C-NMR** (126 MHz, CDCl_3_) *δ* 194.30, 171.67, 165.73 (d, *J* = 255.8 Hz), 143.09, 137.69, 136.61, 130.78, 128.65, 123.40, 119.16, 116.27, 115.22, 62.21, 55.56, 30.17, 25.82. **^19^F-NMR** (471 MHz, CDCl_3_) *δ* -103.92. **HRMS** (ESI) calculated for C_18_H_17_FNO_5_S [M+H]^+^: 362.0857, found: 362.0852.

***4-hydroxy-1,3-dimethyl-3-(tosylmethyl)-3,4-dihydroquinolin-2(1H)-one* (4)**. The product was purified by column chromatography on silica gel (petroleum ether/ethyl acetate = 5:1, **R***_f_* = 0.23), white solid (55 mg, 51%): mp: 96–97 °C. **^1^H-NMR** (500 MHz, CDCl_3_) *δ* 7.85 (d, *J* = 8.3 Hz, 2H), 7.40 (ddd, *J* = 14.9, 7.3, 1.4 Hz, 4H), 7.14 (td, *J* = 7.5, 0.7 Hz, 1H), 7.02 (d, *J* = 8.1 Hz, 1H), 5.20 (d, *J* = 3.0 Hz, 1H), 3.79 (dd, *J* = 14.4, 9.6 Hz, 1H), 3.66 (d, *J* = 14.5 Hz, 2H), 3.35 (s, 3H), 2.47 (s, 3H), 1.41 (s, 3H). **^13^C-NMR** (126 MHz, CDCl_3_) *δ* 171.34, 145.14, 138.50, 137.84, 130.04, 130.00, 129.82, 127.70, 124.55, 123.75, 114.70, 72.01, 58.52, 47.95, 30.14, 21.69, 20.59. **HRMS** (ESI) calculated for C_19_H_22_NO_4_S [M+H]^+^: 360.1264, found: 360.1259.

***3-methyl-3-(tosylmethyl)quinoline-2,4(1H,3H)-dione* (5)**. The product was purified by column chromatography on silica gel (petroleum ether/ethyl acetate = 5:1, **R***_f_* = 0.24), white solid (51 mg, 49%): mp: 141–142 °C. **^1^H-NMR** (500 MHz, CDCl_3_) *δ* 9.70 (s, 1H), 8.04–7.99 (m, 1H), 7.76 (d, *J* = 8.3 Hz, 2H), 7.56–7.52 (m, 1H), 7.29 (d, *J* = 8.1 Hz, 2H), 7.18 (t, *J* = 7.5 Hz, 1H), 6.98 (d, *J* = 8.0 Hz, 1H), 4.26 (dd, *J* = 29.7, 14.1 Hz, 2H), 2.36 (s, 3H), 1.48 (s, 3H). **^13^C-NMR** (126 MHz, CDCl_3_) *δ* 194.31, 173.37, 144.65, 140.74, 138.52, 136.36, 129.72, 128.23, 127.84, 123.70, 117.99, 116.82, 61.91, 55.07, 25.58, 21.59. **HRMS** (ESI) calculated for C_18_H_18_NO_4_S [M+H]^+^: 344.0951, found: 344.0947.

## 4. Conclusions

In summary, we achieved a visible light-promoted cascade sulfonylation/cyclization reaction by using selenosulfonates as sulfonyl-centered radical sources. Various structurally diverse *N*-heterocycles quinoline-2,4-diones were obtained under metal-free/photocatalyst-free conditions. Large-scale synthesis and derivatization study via carbonyl reduction to produce easily modified hydroxyl groups and convenient *N*-Ts deprotection showed the potential utility of this strategy. Through radical-trapping experiments, the rational mechanism is proposed. It is envisaged that this reaction may be useful for the synthesis of more complicated sulfone-containing heterocyclic compounds.

## Data Availability

The data presented in this study are available in the article and Appendix A.

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
