# Peer review of "Visible Light-Induced Cascade Sulfonylation/Cyclization to Produce Quinoline-2,4-Diones under Metal-Free Conditions"

_molecules, 2023, doi:10.3390/molecules28073137_

Round 1

Reviewer 1 Report

Quite an interesting paper suitable for publication in Molecules with minor revisions (listed below)

Corrections required

1.Sulfonylation/Cyclization (tytuł, introduction) sulfonylation cyclization (abstract)

1. Abstract: aryl and alkyl groups are not “functional groups” but rather “substituents”.”

sulfonylation cyclization” should be replaced  by “sulfonylation/cyclization”

2.Scheme 3.”.Ts and.SePh” “ should be replaced by “∙Ts and ∙SePh

3. Scheme 3. In the legend “Proposed Reaction Mechanism” should be replaced by “Proposed reaction mechanism”

4. In “3.1. Materials and Instruments” the Authors write that the NMR spectra were measured on two instruments (500 MHz and 600 MHz for 1H), while all reported data were taken on one apparatus (at 500 MHz).

5. In photochemical experiments, the characteristics of the emission of the light source used (lmax, power, distance from the reactor) and the possibility of absorption of this light by the suggested reagent are particularly important. In my opinion, the Authors should present such data.

Author Response

Responses to Referee 1:

1. Sulfonylation/Cyclization (tytuÅ‚, introduction) sulfonylation cyclization (abstract); Abstract: aryl and alkyl groups are not “functional groups” but rather “substituents”. “sulfonylation cyclization” should be replaced  by “sulfonylation/cyclization”

Answer: Thank you very much for your patient corrections. Appropriate modifications have been made in the revised manuscript.

2. Scheme 3.”.Ts and.SePh” “ should be replaced by “∙Ts and âˆ™SePh”

Answer: Thank you very much for your patient corrections. Appropriate modifications have been made in the revised manuscript.

3. Scheme 3. In the legend “Proposed Reaction Mechanism” should be replaced by “Proposed reaction mechanism”

Answer: Thank you very much for your patient corrections. Appropriate modifications have been made in the revised manuscript.

4. In “3.1. Materials and Instruments” the Authors write that the NMR spectra were measured on two instruments (500 MHz and 600 MHz for 1H), while all reported data were taken on one apparatus (at 500 MHz).

Answer: Thank you very much for your patient corrections. Appropriate modifications have been made in the revised manuscript.

5. In photochemical experiments, the characteristics of the emission of the light source used (lmax, power, distance from the reactor) and the possibility of absorption of this light by the suggested reagent are particularly important. In my opinion, the Authors should present such data.

Answer: Thank you very much for your constructive comments. In this photochemical experiments, we used a 30W blue LED (lmax=455-460 nm), with the reaction bottle 2 cm from the light source. Appropriate description has been added in the revised manuscript.

Reviewer 2 Report

I have reviewed this manuscript carefully; it is clear, well-written, and easily understandable. The manuscript can be published in the “Molecules” after minor revision. The detailed comments are as follows:

 1.     Introduction: Authors should only include closely related references.

  1. Table 1: Optimization: Authors should explain the role of H2O.
  2. Scheme 2: Radical-trapping experiment: Authors should perform a reaction without starting material 1a (with TsSePh and any one radical inhibitor under standard conditions) and report the product(s).
  3. Scheme 3: The mechanism is similar to those previously reported by Ya-Min Li and coworkers1. Furthermore, have the authors tried to confirm/isolate the intermediated C for further confirmation of the proposed mechanism? It could be accessible when you perform the reaction without H2O.
  4. The melting points of these compounds (3a3t3u3w3x, and 3z) must be the same as published by Ya-Min Li and coworkers1 and Xin Wang and Runtao Li and coworkers.2 But unfortunately, they are not. Why?

Compound 3a (mp = 141-142°C), Reported1 160-161°C.

Compound 3t (mp = 139-140°C), Reported2 118-120°C.

Compound 3u (mp = 174-175°C), Reported2 173-175°C. OK

Compound 3w (mp = 89-90°C), Reported2 153-155°C.

Compound 3x (mp = 153-154°C), Reported2 135-137°C. 

Compound 3z (mp = 174-175°C), Reported2 145-147°C. 

Ref. 1: Adv. Synth. Catal. 2016358, 3616-3626; Ref. 2: RSC Adv.20166, 11754-11757.

6.     Authors should provide clean NMR data of compounds 3c (1H and 13C), 3d (1H and 13C), 3e (1H), 3f(13C), 3k (1H and 13C), 3l (1H and 13C), 3m(1H and 13C), 3o(1H),  3u(13C), 4(1H and 13C) and 5(1H) and melting points of all the compounds in their pure forms. Please cross-check the number of carbons and protons in the experimental part.

Author Response

Responses to Referee 2:

I have reviewed this manuscript carefully; it is clear, well-written, and easily understandable. The manuscript can be published in the “Molecules” after minor revision. The detailed comments are as follows:

1. Introduction: Authors should only include closely related references.

Answer: Thank you very much for your patient corrections.

2. Table 1: Optimization: Authors should explain the role of H2O.

Answer: Thank you very much for your constructive comments. Appropriate description has been added in the revised manuscript.

3. Scheme 2:Radical-trapping experiment: Authors should perform a reaction without starting material 1a(with TsSePh and any one radical inhibitor under standard conditions) and report the product(s).

Answer: Thank you very much for your constructive comments. When we performed the radical-trapping experiments without 1a, we could also detect the adducts 6, 8 and even adduct between SePh radical and 1,1-diphenylethylene. All these results indicate that Ts and SePh radicals were generated initially under visible irradiation.

4. Scheme 3:The mechanism is similar to those previously reported by Ya-Min Li and coworkers1. Furthermore, have the authors tried to confirm/isolate the intermediated Cfor further confirmation of the proposed mechanism? It could be accessible when you perform the reaction without H2O.

Answer: Thank you very much for your constructive comments. When the reaction was performed without H2O, we did not get the desired imine C. After HRMS analysis, the product of selenosulfonylation of double bond was detected. 

5. The melting points of these compounds (3a, 3t, 3u, 3w, 3x,and 3z) must be the same as published by Ya-Min Li and coworkers1and Xin Wang and Runtao Li and coworkers.2 But unfortunately, they are not. Why?

Compound 3a (mp = 141-142°C), Reported1 160-161°C.

Compound 3t (mp = 139-140°C), Reported2 118-120°C.

Compound 3u (mp = 174-175°C), Reported2 173-175°C. OK

Compound 3w (mp = 89-90°C), Reported2 153-155°C.

Compound 3x (mp = 153-154°C), Reported2 135-137°C. 

Compound 3z (mp = 174-175°C), Reported2 145-147°C. 

Ref. 1: Adv. Synth. Catal. 2016, 358, 3616-3626; Ref. 2: RSC Adv., 2016, 6, 11754-11757.

Answer: Possible dosage of selected compounds, purity reasons, melting points determined by us differ from those reported. Currently we have completed the purification and melting point determination of the compounds. Because of the epidemic, we sent compounds to other institution, but the NMR characterization did not finish. Considering the approaching return time, we will now submit the revised manuscript. In the later proof stage, we will complete the replacement task of the NMR spectrum.

We also request the editorial department to believe and pay attention to this matter.

6. Authors should provide clean NMR data of compounds 3c(1H and 13C), 3d(1H and 13C), 3e (1H), 3f(13C), 3k (1H and 13C), 3l (1H and 13C), 3m(1H and 13C), 3o(1H),  3u(13C), 4(1H and 13C) and 5(1H) and melting points of all the compounds in their pure forms. Please cross-check the number of carbons and protons in the experimental part.

Answer: Thank you very much for your patient corrections. Currently we have completed the purification and melting point determination of the compounds. Because of the epidemic, we did not finish the NMR characterization. Considering the approaching return time, we will now submit the revised manuscript. In the later proof stage, we will complete the replacement task of the NMR spectrum.